# Setting Priorities to Inform Assessment of Care Homes’ Readiness to Participate in Healthcare Innovation: A Systematic Mapping Review and Consensus Process

**DOI:** 10.3390/ijerph17030987

**Published:** 2020-02-05

**Authors:** Frances Bunn, Claire Goodman, Kirsten Corazzini, Rachel Sharpe, Melanie Handley, Jennifer Lynch, Julienne Meyer, Tom Dening, Adam L Gordon

**Affiliations:** 1Centre for Research in Public Health and Community Care, University of Hertfordshire, College Lane, Hatfield, Hertfordshire AL10 9AB, UK; R.sharpe3@herts.ac.uk (R.S.); M.j.handley@herts.ac.uk (M.H.);; 2University of Maryland School of Nursing, Baltimore, MD 21201, USA; Kcorazzini@umaryland.edu; 3Care for Older People, City, University of London, London EC1V OHB, UK; j.meyer@city.ac.uk; 4Division of Psychiatry and Applied Psychology, University of Nottingham, Nottingham NG7 2TU, UK; tom.dening@nottingham.ac.uk; 5Division of Medical Sciences and Graduate Entry Medicine, University of Nottingham, Derby DE22 3NE, UK; Adam.gordon@nottingham.ac.uk

**Keywords:** long-term care, organisational, context older people, care homes

## Abstract

Organisational context is known to impact on the successful implementation of healthcare initiatives in care homes. We undertook a systematic mapping review to examine whether researchers have considered organisational context when planning, conducting, and reporting the implementation of healthcare innovations in care homes. Review data were mapped against the Alberta Context Tool, which was designed to assess organizational context in care homes. The review included 56 papers. No studies involved a systematic assessment of organisational context prior to implementation, but many provided post hoc explanations of how organisational context affected the success or otherwise of the innovation. Factors identified to explain a lack of success included poor senior staff engagement, non-alignment with care home culture, limited staff capacity to engage, and low levels of participation from health professionals such as general practitioners (GPs). Thirty-five stakeholders participated in workshops to discuss findings and develop questions for assessing care home readiness to participate in innovations. Ten questions were developed to initiate conversations between innovators and care home staff to support research and implementation. This framework can help researchers initiate discussions about health-related innovation. This will begin to address the gap between implementation theory and practice.

## 1. Introduction

In England, there are almost three times as many care home places as there are beds in the acute hospital sector, and one in six people aged 85 or over are living permanently in a care home [1]. Care home residents have complex healthcare needs due to multiple comorbidities (including dementia) yet do not always have access to the healthcare services that they would have if they were living in their own home [2]. In the UK, most care homes do not have registered nursing staff on site [3].

There is a growing recognition of the need for care home specific evidence that informs and improves healthcare of older people in these settings [4,5]. How improvements to healthcare for residents in care homes are implemented depends on a range of factors, including institutional and sectoral priorities, leadership styles, communication patterns, staff interest, and the quality of pre-existing relationships between staff and visiting healthcare professionals [6,7,8]. Understanding from the outset how the organisational context and culture of a care home influences readiness to participate in change is important. It has the potential to shape how healthcare professionals and care home staff plan their work together and explain the variability of uptake of new initiatives across the sector. 

Implementation science recognises that differences in context influence innovation and implementation [9]. Context is a broad concept, and multiple implementation frameworks have operationalised the individual, organisational, and environmental factors involved [10,11]. Despite the rapid growth in implementation science in the healthcare sector in general, there is limited knowledge regarding how context affects innovation and implementation in care homes [12,13]. This is especially relevant when implementation involves practitioners from different types of organisations (public and private, health, and social care), with overlapping but distinct priorities, beliefs, and values.

A Canadian programme of work (Alberta Context Tool© (ACT)) has linked assessment of the organisational context of care with a care home’s capacity to embed new ways of working and caring into its everyday practice [14,15]. This has been done by studying different elements of organisational context, such as leadership styles and communication patterns and their impact on implementation of innovative models of care [16]. The aim of our study was to draw on this work to explore in more detail how the organisational context of the care home, and its constituent elements, might shape care home capacity and readiness to engage alongside health services in innovative approaches to care delivery. We aimed to identify the extent to which researchers have considered organisational contextual factors when planning and reporting the implementation of healthcare interventions in care homes. Organisations involved in innovation in care homes might then use these results to measure and better understand their local care home organisational contexts before attempting to implement changes.

## 2. Methods

The study had two phases. Phase 1 was a systematic mapping review of care home research. A mapping review is based on the concept that published articles not only represent findings but indirectly represent activity related to the finding [17,18]—in this case, the impact of organisational context on implementation. We used the domain headings of the Alberta Context Tool to assess whether care home studies reported any pre or post hoc consideration of the impact of organisational context on readiness to engage in healthcare innovations. Phase 2 consisted of two consensus workshops where findings were reviewed using nominal group technique and stakeholders identified possible questions that practitioners could use to assess care home readiness. 

### 2.1. Phase 1 Systematic Mapping of The Literature

#### 2.1.1. Inclusion Criteria

We included Randomised Controlled Trials (RCTs) and process evaluations in three main areas: telehealth (video consultation and remote monitoring); integrated working between care home staff and visiting healthcare professionals; and comprehensive assessment and care planning, for example by general practitioners (GPs). The reason for including these areas was that they had been selected by commissioners in the English National Health Service (NHS) as priorities for improving care home residents’ contact with the healthcare system. This was part of a broader strategic plan for system-wide change in NHS England [19], where interventions were piloted in six geographically disparate health economies identified as ‘Vanguard sites’ (known as the New Care Model Vanguard sites for Enhanced Health in Care Homes) [20]. 

In addition, we included RCTs, process evaluations, and qualitative studies that had reported on how context had informed uptake and implementation but whose intervention differed from those three areas of care.

#### 2.1.2. Search Strategy

We searched PubMed and CINAHL for records published between 2009 and July 2016. The search was updated in June 2018. The search strategy (see Table 1) involved broad terms based on those used in a previous mapping review of care home research [4]. Non RCTs, such as process evaluations and qualitative studies, were identified from personal knowledge and from lateral searches, such as citation and keyword searches on Google Scholar.

#### 2.1.3. Data Extraction and Analysis

Search results were downloaded into bibliographic software, and duplicates were deleted. Two authors independently screened the first 20 titles and abstracts to check for agreement (R.S., C.R.). The data extracted included information on study aims/research questions, interventions (including how care home staff were involved), participants, setting, type and size of care home, country, and information applicable to organisational context. The focus of data extraction was on factors related to implementation rather than the effectiveness of the intervention. The data was extracted by one of the following authors (R.S., C.R., M.H.) with 20% checked by a second author (F.B.). Disagreements relating to inclusion or data extraction were resolved by discussion with a third author (C.G. or F.B.). 

We mapped the extent to which studies reported on implementation factors outlined in the Alberta Context Tool [21]. The rationale for choosing this particular tool is that it focuses on constructs related to organisation, addresses context assessment in long-term care facilities similar to English care homes, has been used in long-term care research, and draws upon the theoretical work of the Promoting Action on Research Implementation in Health Services (PARIHS) framework [22]. The ACT includes eight dimensions that are comprised of 10 contextual concepts: (1) leadership, (2) culture, (3) evaluation, (4) social capital, (5) structural and electronic resources, (6) formal interactions, (7) informal interactions, (8) organisational slack—staffing, (9) organisational slack—space, and (10) organisational slack—time [14,15]. For the analysis, the data were mapped against the ACT framework to determine if the study assessed care home contextual factors in planning or conducting the study and/or reported on the potential impact of factors on the implementation of the intervention. 

### 2.2. Phase 2 Consensus Workshops

In phase 2, we tested the findings of the review with key stakeholders working in sites that had received additional NHS funding to improve working between healthcare professionals and care homes [20]. We invited care home managers and frontline care home staff, care home researchers, NHS commissioners, and providers of services to care homes to consensus workshops in two areas in England (Nottingham and London). Invitations were sent via the leads for each Vanguard site. Care home staff and managers who responded were self-selecting and were recruited on the basis of their interest in the topic rather than their representativeness. However, to be eligible, participants had to have direct experience working with NHS services and care homes. Before attending the half-day workshop, participants were sent a briefing document outlining the background to the research and the aims of the meeting. To structure the discussion at each workshop and rank the importance of the findings, we used the nominal group technique. This is a process that promotes the generation of ideas and enables the participation of all group members. The process involves four stages: (1) the generation of ideas, (2) recording of ideas, (3) discussing of ideas, and (4) prioritising of ideas [23]. To begin with, review findings were presented using the ACT headings to structure the discussion, and participants were asked whether findings resonated with their experiences. Research team members and participants recorded group discussions on index cards and flipcharts. Initial discussions resulted in 21 characteristics likely to affect care home readiness to participate in NHS-led service improvements. Using the nominal group technique [23], participants ranked these in order of importance. Thirty-five participants attended the two workshops (Table 2). Findings from the two phases were synthesised by members of the team (C.G., F.B., R.S., J.L., A.G.), and the key themes or ideas were expressed as questions. These questions were developed by the researchers and were not tested further with participants.

## 3. Results

### 3.1. Systematic Mapping

Fifty-six papers from 48 studies met our inclusion criteria (Figure 1). These included 36 RCTs [24,25,26,27,28,29,30,31,32,33,34,35,36,37,38,39,40,41,42,43,44,45,46,47,48,49,50,51,52,53,54,55,56,57,58,59] and 20 process evaluations or qualitative explorations of implementation [60,61,62,63,64,65,66,67,68,69,70,71,72,73,74,75,76,77,78,79]. We found studies from 10 countries: the United Kingdom (13), the United States (11), Australia (8), the Netherlands (5), Norway (3), Belgium, New Zealand, Canada (all 2), China (1), and France (1). An overview of the types of interventions and the domains of the ACT covered can be seen in Table 3. For further details of individual studies, see Appendix A.

None of the papers included a structured and comprehensive assessment of context that considered all the domains of the ACT. However, one study [48] used the PRECIS-2 tool [80] to evaluate implementation of their pragmatic randomised controlled trial, and another used the findings of an RCT in nursing homes to develop a tool to assess and manage the challenges facing complex organisational interventions [74]. However, the latter framework focuses on issues that arise during a study rather than anticipating and planning for them in the study design and set-up.

The most commonly considered areas of organisational context were leadership, culture, formal interactions, and staff availability. Where contextual assessment did feature, it was most frequently used as an unstructured post-hoc exploration of why an intervention had or had not worked. Detailed mapping of individual studies against the domains are shown in Appendix A. 

#### 3.1.1. Leadership

Most papers reviewed (n=40) noted that care home leadership influenced the uptake of a given innovation. Studies cited the importance of leadership from care home managers and/or senior direct-care staff [29,38,53,54,64,77] and from staff acting as clinical champions or persuasive leaders. Some studies attempted to address leadership issues prospectively through early engagement with care home managers [30,34,36,38,45,66,77] and/or the appointment of clinical champions [26,29,44,45,48,49,50,63,64]. However, in many cases, engagement appeared superficial or limited. Examples where more sustained engagement was built into the trial design include the MARQUE study [45] and the WHELD study [44,64]. In the former, researchers held regular supervision and troubleshooting meetings with care home managers, and in the latter, champions were given sustained support and coaching aimed at building their confidence. This engagement process appeared to contribute to sustained delivery of the intervention.

A number of studies focused almost exclusively on the negative impact of a lack of leadership [38,42,43,53,68,72,78]. Issues cited included poor role clarity, managers’ resistance to change, delegation of responsibilities to staff without the skill or authority to implement change, turnover of managers, and insufficient management attention to the innovation. Whilst these are well known to affect implementation, they had generally not been addressed as part of the innovation’s development.

#### 3.1.2. Culture

Positive cultural attributes were identified as those factors that gave time and resources to staff education, reinforcement of learning, and quality improvement [43,45,52,54,64,65,66,67]. Feedback on progress encouraged a sense of shared ownership of a given change [53]. Uptake was more likely when an intervention was acceptable to healthcare professionals, residents, and staff; when it fitted with existing care home routines; and when there were opportunities for ongoing consultation with staff [36,45,60,69,70]. Some studies attempted to proactively address the impact of culture, for example by involving stakeholders, such as care homes staff, in the development of the intervention [36,63,74].

Culture negatively affected uptake when the systems of care and required staffing levels were incompatible with those proposed by healthcare professionals or if care home staff felt that the proposed change implied a criticism of current practices [24,38,55,61]. Practices within the care home that specifically worked against the successful uptake of initiatives were the following: a task-focused approach to care, a preoccupation with risk reduction, or staff with a limited skill set working with residents who had advanced dementia [34,37,62,68,71,73,75]. Studies highlighted the importance of managerial support to change perceptions about what constituted real work. For example, supporting people so that they felt comfortable to sit and talk with residents rather than engaging in task-based care [49].

#### 3.1.3. Evaluation

Care homes’ use of data to assess performance and achieve outcomes was discussed in terms of staff’s familiarity with gathering data and how they used information to inform quality improvement, specifically whether care homes easily provide information about residents’ characteristics, document their participation and health-related outcomes, or provide information about relatives’ involvement in care [30,54,60,71,72,73,76]. One study noted the related challenges of synthesising data from the multiple data sources within a care home [55]. Other studies described the benefits of engaging in pre-intervention work or adapting to current processes to ensure consistent documentation of care [40,79].

#### 3.1.4. Social Capital

Social capital recognises the existing resources and support networks, both formal and informal, available to a care home to deliver care. It helps to explain why care homes with similar populations may be more or less resilient and responsive to change. This domain was not explored in most of the research reviewed, although some studies reflected on how homes’ connections, particularly with external services, affected implementation of care initiatives [27,31,69,70,72,76]. Specifically, absent or poor connections with general practitioners (community physicians), secondary care (hospitals), and professional or academic organisations were important in affecting how care homes worked with visiting healthcare practitioners. Two studies reported on the advantages of having clinicians working with care homes to support interventions to improve the quality of care [25,36].

#### 3.1.5. Informal and Formal Interactions

Low GP participation or resistance from GPs [29,34,47,61,63,69,77,78,79,81], limited opportunities for formal communications in multidisciplinary team meetings [27,28,60,72,73], and poor communication within the organisation [55,68,73] were factors reported to be barriers to implementation. One study recognised the need in future work to address how group dynamics and peer pressure facilitates (or not) the adoption of the intervention and the possible benefits of preparatory coaching to build staff confidence in their practices [30]. Indeed, approaches that fostered a combination of formal and informal interactions, for example through regular meetings, coaching, interactive approaches, and good communication of information, were all identified as facilitators [30,38,52,67,77]. This included interactions between care home staff, between staff and external health professionals, and between staff and researchers.

#### 3.1.6. Structural/Electronic Resources 

Studies showed that when care homes experienced system changes or reorganisation, uptake of new interventions was limited [25]. Some studies did acknowledge the impact of this; for example, one study excluded care homes in which major innovation projects had recently been implemented [30]. Care home record keeping systems, and limited access to computers could have a negative impact on the collection of outcome data or participation in the study [49,51,53,72]. Some studies provided support or training to staff to improve recordkeeping or the use of IT in the care home [48,64,66]. In the WHELD study, recordkeeping for the research influenced the wider practice of the care home and led to an overall improvement in documentation [64]. This appeared to be because staff were actively engaged in the project and because the research enabled them to see the connection between care home practice and outcomes [64].

#### 3.1.7. Organisational Slack

The biggest issue, identified by almost all studies (n = 43), was staff availability and capacity. This was expressed in four ways: staff workload, staff turnover, staff skills, and whether an innovation was seen as a priority by the care home. Some studies suggested that funding for staff time to deliver an intervention or attend training might ease the problem of staff availability [29,42,45,53,63,68,75]. Making time to build relationships, to agree how to work together, and to establish if the intervention was relevant to the care home was also important [27,41,56,59,60,64]. Space was less frequently mentioned as an issue, although two studies reported that the physical structure of the home could be challenging [68,71], for example making it difficult to maintain residents’ privacy during research interviews [71].

### 3.2. Consensus Workshops

Participants at the consensus workshops recognised and validated the review findings. They found it challenging, however, to articulate how they would quantify these different aspects of care home context or assess how they affected readiness to participate, implement, and sustain innovation. For example, participants endorsed the importance of having enough time to get to know each other to build a shared agenda and mutually beneficial working relationships but were unsure how this is achieved or how many resources are required.

Similarly, everyone identified leadership approaches in the care home as key. They struggled, however, to unpack what type of leader was important, whether the level of managerial involvement was significant, or how staff turnover and availability might affect uptake of the interventions. Fewer still, despite personal experience in instituting healthcare change in and with care homes, had considered how the internal systems of the care home and surrounding networks of care affected uptake or whether the priorities of the NHS always aligned with those of the care homes. 

Evidence showing that it was frequently difficult to engage healthcare professionals with different care home innovations resonated with participants at both workshops. Participants also noted that there was little information on what was required to ensure that visiting healthcare professionals were ready to work with care homes and limited evidence about how healthcare practitioners’ prior knowledge and experience of working in care homes affected implementation. They observed that studies seldom considered if an innovation was wanted or needed by the care home and that they usually failed to partner with care home staff in planning innovation from the outset. 

Workshop discussions identified 21 characteristics likely to affect care home readiness. When asked to rank them, participants identified the following as most important: having a capable and confident manager, alignment of priorities and staff buy-in, engagement with a care quality vision, evidence of a culture of change, and receptiveness of manager and senior staff to engage in change. 

The findings were then developed by the research team into a series of questions (Table 4). These questions were designed to provide a framework to promote conversations between researchers, practitioners, and commissioners when considering innovation in care homes.

## 4. Discussion

To identify and map the contextual influences that affect successful implementation of healthcare interventions in care homes, we conducted a review of 56 care home papers and undertook workshops involving 35 participants. Both the review and the workshops offered evidence of how context influenced implementation but less knowledge of how to achieve context ‘readiness’. The most commonly cited components of organisational context were leadership, care home culture, formal interactions, and organisational slack. The review demonstrated that researchers are aware of how context affects uptake of healthcare innovation in care home settings. Despite this, organisational context was generally used to explain problems with implementation post hoc, rather than being planned pre hoc as part of the study design.

Leadership, care home culture, and staff capacity to engage in and prioritise innovations are well recognised as important influences on uptake of innovation [82,83]. Given this, it is disappointing that, in many studies, these well-documented challenges are presented as findings, rather than being used to inform the study design. Overall, there were little data on what is required prior to innovation in terms of research design, involvement of staff, and the resources required. More recent trials in care homes [44,45] have provided accounts of how care home context was addressed in setting up the study and delivering the intervention. For example, the WHELD study highlighted the importance of understanding the experiences of care home staff, engaging them as active members of the research team, and supporting them to develop skills and take ownership of the intervention [64,65]. 

A clear message from the workshops, and from the review, was that it takes time to develop relationships that support effective collaborations between visiting healthcare professionals and care homes. These findings are supported by a recent realist evaluation of health service delivery to care home residents in the English NHS [84]. This study found that service integration between the NHS and care homes was dependent on time, and support from commissioners to develop relational working. It is also suggested that a long preparatory period, consultation with a range of frontline staff and care home residents [83], and co-design of interventions could improve implementation [85].

Very few studies provided the opportunity for care home staff to comment on the relevance of research or the experience of participation. Further, residents’ and relatives’ voices are also largely missing from this study. This is, in part, because they were absent in the papers reviewed. It is a significant limitation that so little can be said about how residents and their representatives influence the planning and uptake of healthcare interventions.

## 5. Limitations

It is possible that the time limits incorporated in our search strategy meant that we missed relevant studies on care home context. In addition, we did not include study protocols, which might have provided more information about researchers’ plans to address organisational context. Previous mapping reviews of trials conducted in care homes have not, however, revealed significant volumes of implementation literature in the sector that would have been likely to change our findings [4]. The ACT framework provided a useful tool for mapping organisational context. However, we found some overlaps between domains, for example organisational slack around staff and time. We conducted only two workshops with self-selected participants; thus, the consensus rankings need further refinement and testing with a wider audience. 

## 6. Conclusions

Care home context and readiness for change is an important factor for the successful implementation of healthcare initiatives. Approaches to measuring context, such as the ACT, have been applied to individual research studies. Drawing on the findings of this article, there is a clear case to persevere with structural assessment of care home context in research. This requires a systematic approach to assessment at the beginning of an innovation or intervention. This synthesis and stakeholder engagement led to the development of questions that can be used to help researchers, practitioners, and commissioners begin conversations about the measures needed to bring care homes to a state of readiness for successful implementation of healthcare initiatives. Ultimately, asking these questions consistently across diverse care home settings will inform the development of a shared, core set of context assessment tools to support healthcare interventions in care homes in England and elsewhere. 

## Figures and Tables

**Figure 1 ijerph-17-00987-f001:**
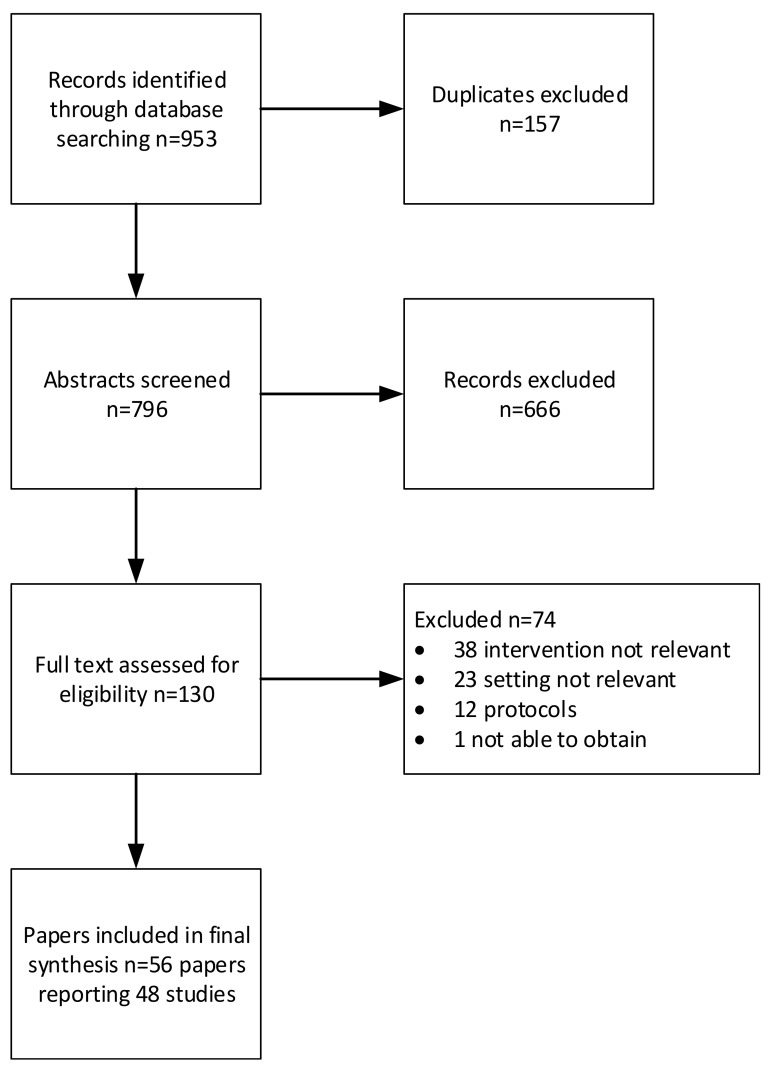
Selection of studies for inclusion in the mapping review.

**Table 1 ijerph-17-00987-t001:** Search terms for the mapping review.

PubMed.
“nursing home” OR “residential facilities” OR “homes for the aged” (MESH) OR nursing homes (TI/AB], care home [TI/AB] OR residential care [TI/AB]
AND “randomised controlled trial” OR “randomized controlled trial” (MESH)
**CINAHL**
“nursing homes” OR “residential facilities” OR “skilled nursing facilities”

**Table 2 ijerph-17-00987-t002:** Workshop participants.

Workshop	Participants	
London workshop n=21 + 3 facilitators	Care home manager Care home representative organisation/charity NHS physician/nurse/therapist working with care homes NHS manager/commissioner Care home researchers	3 4 6 3 5
Nottingham n=14 + 2 facilitators	Care home manager Care home representative organisation/charity NHS physician/nurse/therapist working with care homes NHS commissioner/manager Care home researchers	1 3 4 4 2
Total participants		35

**Table 3 ijerph-17-00987-t003:** Types of interventions and frequency with which Alberta Context Tool (ACT) domains were considered.

Type of Intervention	N of Papers	Citations
Integrated working between care home staff and visiting health professionals	24	[24,25,27,28,32,35,36,38,39,42,43,44,45,47,50,58,60,63,64,65,66,69,78,79]
Comprehensive assessment and care planning	8	[44,45,46,48,49,50,51,64,65,66,77]
Telehealth	3	[31,33,41]
Other type of intervention (but provides detail on organisational context)	27	[26,28,32,33,34,35,38,39,44,47,48,49,54,58,60,61,62,67,68,70,71,72,73,75]
**Domain of ACT**	**Considered in paper (n=)**	**Citations**
Leadership	40	[24,25,27,28,29,30,31,34,36,38,40,42,43,44,45,46,47,48,49,50,52,53,54,55,56,58,59,60,62,63,64,66,67,68,69,72,73,75,77,78]
Culture	32	[24,27,34,36,37,42,45,46,48,49,52,53,54,55,58,59,62,63,64,66,67,68,69,70,72,73,74,75,76,78]
Evaluation	12	[25,39,40,44,45,46,48,49,50,51,64,66]
Social capital	13	[25,31,49,59,64,65,66,68,69,70,72,76,77]
Informal interactions	11	[30,49,55,57,59,60,64,65,67,72,77]
Formal interactions	30	[27,28,29,30,31,34,38,42,45,46,48,49,50,51,52,55,61,63,64,65,66,68,69,70,72,73,76,77,78,79]
Structural/electronic resources	22	[25,30,31,41,42,44,45,46,48,49,50,51,52,53,55,57,61,64,65,66,72,77]
Organisational slack—staff	36	[24,26,27,29,30,34,36,38,42,45,47,48,49,50,52,53,54,56,57,58,59,60,61,63,64,65,66,67,68,73,74,75,76,77,78]
Organisational slack—space	6	[33,51,66,68,70,71]
Organisational slack—time	30	[24,27,28,29,30,34,37,38,39,45,47,49,50,51,52,53,55,56,58,59,61,65,66,68,70,71,72,73,77,78]

**Table 4 ijerph-17-00987-t004:** Set of questions combining review findings with the workshop priorities.

	Set of Questions Combining Review Findings with the Workshop Priorities
1	Does this intervention align with care home priorities?
2	What evidence is there of senior management interest and enthusiasm for this intervention at the organisational and unit level? Are they willing and able to take an ongoing leadership role in supporting the proposed change?
3	Do care home staff have enough ‘slack and flexibility’ to accommodate the change into their current workload? Will it be recognised as core to their work?
4	How is change discussed (formally and informally) in the care home setting? Who needs to be involved in decision-making about what is being proposed and how it is implemented?
5	What are the recent changes or health-related projects this care home has been involved with?
6	Is there a champion in both the care home and in the linked NHS service with protected time to facilitate change?
7	What are the pre-existing relationships between NHS services and care home staff and networks of care and support around the care home (e.g., general practitioners (GPs), visiting specialists, links with local hospital)?
8	Could the intervention appear judgmental by signalling in a negative way that the care home needs to change?
9	How well do existing care home training programmes and work schedules fit with what is proposed?
10	Will care home staff have to collect and enter new data or are they held in existing systems?

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
