# Peer review of "Setting Priorities to Inform Assessment of Care Homes’ Readiness to Participate in Healthcare Innovation: A Systematic Mapping Review and Consensus Process"

_ijerph, 2020, doi:10.3390/ijerph17030987_

Round 1

Reviewer 1 Report

The described research helps to understand and facilitate the real-world application of implementation science in care home healthcare innovation programs. The manuscript is well written, I propose only a few minor modifications:

1) Although the authors were interested in the assessment of care home contextual factors in the planning phase and later stated that "it is disappointing that in many studies these well documented challenges are presented as findings, rather than being used to inform the study design", they excluded all study protocols from their systematic mapping review. Inclusion of study protocols in the analysis, or justification of their exclusion is proposed.

2) It is suggested to indicate the number of papers excluded in the title-abstract review phase by the specific exclusion criteria (Figure 1).

3) In Phase II, relevant stakeholders were invited from all care homes that had received additional NHS funding for related programs in two areas in England. As a measure of workshop sample representativeness, please report the number of care homes invited, and the number of care homes that accepted the invitation to the workshops.

4) Results on "organizational slack" does not show results on the space context - please supplement this section accordingly. Moreover, authority to initiate change may better fit to the leadership domain.

5) Lines 331-335 to be moved to the Limitations section. 

Reviewer 2 Report

The paper entitled “Setting priorities to inform assessment of care homes’ readiness to participate in healthcare innovation: a systematic mapping review and consensus process” is a properly planned and well-designed study. Moreover, the data included could be potentially important for the developing the innovative solutions, in field of home care medical help.  This publication draws to explore how the organisational context of the care home, and its constituent elements, might shape care home capacity and readiness to engage alongside health services in innovative approaches to care delivery.

What is particularly important? The authors of the manuscript tried to identify the extent to which researchers have considered organizational, contextual factors when planning and reporting the implementation of health care interventions, in home care.

Authors however, did not explain convincingly the choice of the Alberta Context Tool © (ACT) implementation.

Moreover, the authors could also, based on the cited literary data (if possible), specify the age range of patients under the home care service.

The conclusions take the form of a summary of results.

The authors could, at least, try to suggest the solutions of the improvement of the current home care service.

Last but not least, only 41.86 % of cited articles have been published in the last five years.

In the light of the dynamically changing situation of health care, in particular its permanent underfunding, and the aging of the population, it may not lead to the proposal of the appropriate conclusions.

Reviewer 3 Report

After article valuation “Setting priorities to inform assessment of care homes’ readiness to participate in healthcare innovation: a systematic mapping review and consensus process”, I can say that in general it is a pertinent study that in general is well carried out. Nevertheless, the following suggestions are made that may help to clarify the study and the understanding of its development.

At data extraction and analysis section, in 2.2 Phase 2 Consensus Workshops (line 146-151), it should be indicated the data analysis. The data were analysed using qualitative methods and has to indicate how it is carried out and data processing. It is needed complete this section.

In the same way, it is necessary to complete the results section (Consensus workshops: line 274-300).

This section sets out the findings in general but I think I lack you to better specify the opinion of the participants. For example, line 281 "everyone identified leadership approaches in the care home as key". At this point, the authors make it clear that they all agreed, however, we do not know how the other results they indicate were reached.

Is it possible that there are dissonances in the opinions of the different professionals?

I recommend include more detailed information on this aspect.

This synthesis and stakeholder engagement, led to the development of questions that can be used to help researchers, practitioners and commissioners begin conversations about the measures needed to bring care homes to a state of readiness for successful implementation of healthcare initiatives. However, I miss an explanation of how consensus has been reached.

Round 2

Reviewer 3 Report

I think the changes are appropriate